# Enhancing Facial Expression Recognition through Light Field Cameras

**DOI:** 10.3390/s24175724

**Published:** 2024-09-03

**Authors:** Sabrine Djedjiga Oucherif, Mohamad Motasem Nawaf, Jean-Marc Boï, Lionel Nicod, Elodie Mallor, Séverine Dubuisson, Djamal Merad

**Affiliations:** 1Institut de Mathématiques de Marseille (IMM), CNRS, Aix-Marseille University, 13009 Marseille, France; 2Laboratoire d’Informatique et des Systèmes (LIS), CNRS, Aix-Marseille University, 13009 Marseille, Franceseverine.dubuisson@lis-lab.fr (S.D.); djamal.merad@lis-lab.fr (D.M.); 3Centre d’Etudes et de Recherche en Gestion d’Aix-Marseille (CERGAM), Aix-Marseille University, 13013 Marseille, France

**Keywords:** light field cameras, facial expression recognition, multimodality

## Abstract

In this paper, we study facial expression recognition (FER) using three modalities obtained from a light field camera: sub-aperture (SA), depth map, and all-in-focus (AiF) images. Our objective is to construct a more comprehensive and effective FER system by investigating multimodal fusion strategies. For this purpose, we employ EfficientNetV2-S, pre-trained on AffectNet, as our primary convolutional neural network. This model, combined with a BiGRU, is used to process SA images. We evaluate various fusion techniques at both decision and feature levels to assess their effectiveness in enhancing FER accuracy. Our findings show that the model using SA images surpasses state-of-the-art performance, achieving 88.13% ± 7.42% accuracy under the subject-specific evaluation protocol and 91.88% ± 3.25% under the subject-independent evaluation protocol. These results highlight our model’s potential in enhancing FER accuracy and robustness, outperforming existing methods. Furthermore, our multimodal fusion approach, integrating SA, AiF, and depth images, demonstrates substantial improvements over unimodal models. The decision-level fusion strategy, particularly using average weights, proved most effective, achieving 90.13% ± 4.95% accuracy under the subject-specific evaluation protocol and 93.33% ± 4.92% under the subject-independent evaluation protocol. This approach leverages the complementary strengths of each modality, resulting in a more comprehensive and accurate FER system.

## 1. Introduction

In affective computing, methodologies can be divided into two main approaches: unimodal affect recognition and multimodal affective analysis. The former focuses on analyzing a single type of data, such as facial expressions, voice intonation, or physiological signals, while the latter integrates various data sources to gain a more comprehensive understanding of emotional states [1].

Although multimodal affective analysis represents a more robust and efficient system, it faces challenges such as the complexity of synchronizing and integrating data from multiple sources. Additionally, the use of several pieces of equipment can be intrusive, potentially influencing the natural behavior and emotional responses of subjects [2]. In this context, light field technology emerges as a promising solution.

Light field (LF) cameras, also presented as a plenoptic cameras, have the capability to passively capture synchronized multi-view facial structures, which implicitly include depth information. The imaging system’s architecture features a unique setup with a micro-lens array positioned between the primary lens and the image sensor. This configuration allows for the simultaneous capture of both the intensity (spatial information) and direction (angular information) of light in a scene [3].

As light passes through the array of micro-lenses, it follows a predetermined path. Each micro-lens acts as a directional window, capturing light from various directions originating from a 3D point within the scene, thereby creating a micro-image. These collective micro-images, obtained from the entire array, enable the extraction of three distinct types of data: sub-aperture, all-in-focus, and depth map images, as illustrated in Figure 1. This feature is crucial for acquiring multi-directional light information, which facilitates subsequent three-dimensional reconstruction.

In this article, we explore the potential of multimodal fusion in facial expression recognition (FER). This fusion is pivotal in our study, as it integrates varied facial structure details embedded in both RGB and depth map images, which are essential for FER. Our approach emphasizes multimodal fusion as a vital aspect, leveraging comprehensive data to significantly enhance the recognition process. The integration of all modalities obtained with an LF camera is a relatively unexplored territory in research.

To further clarify, we provide a detailed list of the data provided by the light field camera:**Sub-aperture images.** Each sub-aperture (SA) image is formed by extracting pixels at a specific position from each micro-image (see Figure 2). This method ensures that each sub-aperture image captures directional information from the entire micro-lens array after the decoding process, contributing to a comprehensive representation of the scene’s multidirectional characteristics. The dimensions of each SA image, namely, its height and width, depend on the number of micro-lenses. For instance, with the Lytro Illum camera, it is possible to obtain a 15×15 matrix of SA images, totaling 225 SA images (see [5]). Each SA image has 434×625 pixels, representing the number of micro-lenses available in the Lytro camera.**All-in-focus (AiF) image.** Also known as the total focus image, is obtained by combining information from the SA images. This creates a visual representation of the scene where all elements, regardless of their distance from the camera, are sharply focused. Notably, this process emphasizes high resolution [6].**Depth map.** Parallax calculation in light field cameras involves matching SA images to measure the apparent displacement of points viewed from different positions. These data are then used to generate a depth map, indicating the distance of each scene point, relative to the camera, and corresponding to points in 3D space. Advanced algorithms are subsequently applied to refine these depth maps, particularly in areas with minimal or challenging parallax, thus enhancing the accuracy and precision of the measurements [7].

With the aim of studying the benefits of light field cameras in facial expression recognition, we first review existing works in this context in Section 2. Subsequently, in Section 3, we elaborate on the preparation and processing stages and discuss the datasets employed in this study. Following this, we present our model and the results obtained in Section 4 and Section 5, respectively. Finally, our study concludes with a discussion and future perspectives in Section 6.

## 2. Related Work

Advances in light field camera technology have opened new perspectives in the field of facial expression recognition (FER). Recent studies have delved into various methods leveraging the capabilities of these cameras, including extracting detailed depth information and providing multiple perspectives of a scene.

One of the earlier studies by Shen et al. [8] generated a private dataset containing both depth maps and AiF images, capturing the six types of basic emotions and the neutral state identified by Ekman [9]. The face area was cropped, and both the AiF image and depth map were resized to 128×128 and 64×64 pixels, respectively. Subsequently, the Histogram of Oriented Gradients was used for feature extraction, and a Support Vector Machine was employed for FER. The study revealed that using only the AiF image resulted in an average precision score of 45.55%. When the depth map was added, the precision increased to 47.16%.

Research conducted by Sepas-Moghaddam et al. [10,11] introduced a novel approach for FER using SA images from the Light Field Face Database (LFFD). The SA images, acquired along the central vertical and horizontal lines of the sub-aperture mosaic, were resized to 224×244 pixels. These images were fed into two VGG-Face models combined with a Bidirectional Long Short-Term Memory (LSTM) network [12,13]. Both models were enhanced with an attention mechanism prior to classification. A fusion process was employed to generate a final score. A comparative analysis among Neutrality, Angry, Happiness, and Surprise expressions was conducted using the LFFD. This approach provided an accuracy of 87.62%±5.41% under the subject-specific evaluation protocol and 80.37%±9.03% under the subject-independent evaluation protocol.

The same authors extended their work by developing a second model based on the CapsField framework [14]. This model incorporated two CNN sub-networks based on VGG-16 [15] and ResNet-50 [16] architectures to extract spatial features from horizontal and vertical SA sequences. By introducing a capsule network [17] to capture intricate relationships among these features, the convolution operation is omitted in the primary capsule layer. The classification process is executed through two independent dense layers with a softmax activation, merging scores to predict final labels for FER. The comparison focused on Neutral expressions and random facial expressions provided by Light Field Faces in the Wild (LFFW). The results showed 100% accuracy for each category using indoor data for training and outdoor data for testing. Additionally, when the training and testing settings were inverted, accuracies were 90.56% for Neutral and 93.71% for random facial expressions.

In our previous research [18], we explored a variety of models using EfficientNetV2-S [19] along with different recurrent neural networks (RNNs) such as LSTM [20], gated recurrent unit (GRU) [21], bi-directional (Bi) GRU, and Bi-LSTM. The combination of CNN and RNN allows for the extraction of both spatial and angular information from SA images, providing a more comprehensive and accurate analysis of facial expressions. Notably, two configurations achieved higher accuracies. The first one paired a dual-branch EfficientNetV2-S with LSTM, and the second used a single-branch EfficientNetV2-S with Bi-LSTM. These configurations were rigorously tested using exclusively diagonal images from the SA image mosaic, resized to 60×60 pixels from the LFFD dataset. Notably, the dual-branch configuration with LSTM was particularly effective, achieving an impressive average precision score of 82.88%±6.47%.

In this paper, we focus on exploring FER using three modalities obtained from the Light Field Face Database (LFFD): SA images, depth maps, and AiF images. Our objective is to construct a more comprehensive and effective FER system by investigating multimodal fusion strategies.

In the field of affective computing, multimodal fusion involves integrating various data modalities, including images, videos, audio, and bio-electrical signals such as EEG or ECG [22,23,24,25]. Our study specifically delves into the nuances of multimodal fusion by examining the intricate details captured by the LF camera. By leveraging this approach, we aim to uncover complex patterns within these diverse data sources, thereby enhancing our understanding and improving the capabilities of FER systems.

The study of LF images, by exploiting all the information they provide, remains a largely unexplored area. Our research seeks to bridge this gap by thoroughly analyzing the rich data from LF images, which could lead to significant advancements in the field of FER.

## 3. Pre-Training Dataset Selection and Model Configuration for FER

In this section, we outline the steps involved in preparing our model and processing the datasets used for FER. We present the datasets used for pre-training our model. The model obtaining the best score on AiF images is used to synthesize depth maps. This approach aims to develop a robust model capable of recognizing facial expressions using both RGB and depth images.

### 3.1. FER Datasets

For our study, we employed EfficientNetV2-S [26], a CNN known for its performance, efficiency, and scalability in image classification tasks. This CNN was pre-trained on multiple datasets specifically designed for FER to provide a broad spectrum of emotional data. This initial pre-training aims to enhance the accuracy and effectiveness of our FER models.

The datasets used for pre-training include the following:**FER2013:** The Facial Expression Recognition 2013 (FER2013) dataset [27] was specifically proposed for FER research. It comprises 35,887 facial images, each one classified into one of seven distinct expressions: Anger, Disgust, Fear, Happiness, Sadness, Surprise, and Neutrality. Each grayscale image is 48×48 pixels, providing a consistent format for analysis and model training.**CK+48:** The Extended Cohn-Kanade (CK+48) dataset [28] was also proposed for FER research. It contains 980 grayscale 48×48 pixel images. The dataset covers seven distinct facial expression categories: Fear, Disgust, Sadness, Happiness, Neutrality, Surprise, and Anger, offering a comprehensive array of expressions for analysis.**AffectNet:** The AffectNet dataset [29] contains 291,650 facial images, a blend of RGB and grayscale, all standardized to a size of 224×224 pixels. Each image is manually annotated by 12 experts. These annotations encompass eight facial expressions: Neutrality, Happiness, Anger, Sadness, Fear, Surprise, Disgust, and Contempt. Additionally, the dataset provides annotations for the intensity of valence and arousal. This wealth of data makes AffectNet a comprehensive tool for studying facial expressions and affect in naturalistic settings.

Figure 3 presents a sample image for each facial expression from these datasets, providing a visual representation of the data types used in our analysis.

Using these specified datasets on EfficientNetV2-S yielded notable accuracies: **66.51%** for FER2013, **94.87%** for CK+48, and **76.46%** for AffectNet.

### 3.2. Depth Map for FER Dataset

Since there is no existing dataset for FER with depth maps, we synthesize these images from the dataset that yields the best performance on AiF images. We consider that AiF images are similar to RGB images obtained from standard cameras in our approach.

To create these depth maps, we use the Depth Anything model [30], enabling us to generate depth information for facial expression images. The DAM is a recently released foundation model for monocular depth estimation (MDE). It is based on a vision transformer architecture and has been trained on a vast dataset of approximately 63.5 million images, including 1.5 million labeled and 62 million unlabeled images. This model leverages advanced deep learning techniques to accurately predict depth information from 2D images, making it highly effective for generating high-quality depth maps that capture intricate details and variations in facial structures. By incorporating these synthesized depth maps, we aim to enhance our facial expression recognition system, providing a richer and more comprehensive understanding of emotional expressions.

Figure 4 illustrates the depth images synthesized from the RGB images of the AffectNet dataset.

## 4. Detailed Methodology and Implementation of the Proposed Approach

In this section, we provide a detailed exploration of the architectures developed for handling SA images, depth maps, and AiF images. We describe the designs of both our unimodal and multimodal architectures.

Studying unimodal and multimodal approaches is essential for understanding the benefits of the data obtained from the LF camera in enhancing FER performance. Additionally, we discuss the hyperparameters used in our model, their importance in improving its effectiveness, and the evaluation protocols we have implemented.

### 4.1. Unimodal Approaches

In order to study facial expression recognition from AiF images and depth maps separately, our model employs EfficientNetV2-S as the CNN backbone. This backbone is augmented by two successive blocks comprising dense layers, batch normalization, and dropout layers. The dense layers fully connect the network, allowing it to learn complex patterns in the data. Batch normalization layers stabilize and accelerate the training process by normalizing the input to each layer, thereby improving the model’s performance and convergence. Dropout layers prevent overfitting by randomly setting a fraction of input units to zero during training, which helps the model generalize better to new data. The architecture concludes with a classification layer using a softmax function to output the probabilities of each facial expression class.

The architectural details of this model are illustrated in Figure 5. We refer to the model used for AiF images as “Model_1” and the model used for depth images as “Model_2”. Both models share the same architecture.

For the model dedicated to SA images, referred to as “Model_3”, we use EfficientNetV2-S to extract spatial information, similar to the approach used for AiF images and depth maps. Following this, BiGRU is employed to extract angular information from the SA images. Next, two stacks of dense, batch normalization, and dropout layers are applied. For the first stack, each layer is wrapped in a TimeDistributed layer to handle the angular aspect of the data. An attention mechanism is integrated between these two stacks to enhance the model’s focus on the most relevant features by fusing the angular and spatial information vectors. The attention layer calculates alignment scores for each time step of the input sequence, converts these scores into attention weights using a softmax function, and produces a context vector that captures the most pertinent information from the input sequence. This allows the model to emphasize the important parts of the input, improving its ability to recognize facial expressions accurately. The model concludes with a classification layer using a softmax function.

The detailed architecture of “Model_3”, dedicated to SA images, is depicted in Figure 6.

### 4.2. Data Fusion Strategies

A key feature of our method is the fusion layer tailored for light field camera data, treating AiF, SA, and depth images as separate modalities.

We explored two fusion strategies: decision level and feature level. Decision-level fusion combines final model outputs, boosting accuracy and reliability by leveraging strengths of different classifiers. In contrast, feature-level fusion integrates the data from diverse modalities before classification, enriching the feature set for a more holistic analysis (see Figure 7).

This setup aims to investigate both fusion types within our framework, potentially enhancing the robustness and accuracy of facial expression recognition. By comparing these strategies, we seek to identify the optimal approach for integrating data from multiple modalities, thereby improving the performance and reliability of our emotion recognition models. Our evaluation of the best fusion technique will specifically focus on the integration of AiF with SA images.

### 4.3. Multimodal Approaches

Given our interest in exploring the contribution of LF cameras to FER, and recognizing that the data provided by these cameras come in three distinct types, we aimed to understand the contribution of each type individually, as well as in combinations of two modalities and all three together. To investigate this, we compared pairs of models at a time, then all three together, and merged them using an averaging layer.

Model_4 represents the fusion of the model for SA images with the model for depth images. Model_5, on the other hand, is the fusion of the model for SA images with the model for AiF images. Model_6 combines the models for AiF and depth images. Finally, Model_7 integrates all three modalities.

### 4.4. Light Field Camera Dataset

The IST-EURECOM Light Field Face Database (IST-EURECOM LFFD) [4] stands as a unique resource for FER research using an LF camera, featuring 100 subjects each captured with a Lytro Illum camera. This extensive database, with two sessions per subject and 20 samples each, encompasses a range of facial expressions, activities, poses, lighting conditions, and occlusions. It include raw LF images, AiF images, depth maps, and detailed metadata.

In our study, we specifically analyze the facial expressions available in the dataset. It contains only three of the six basic emotions—anger, joy, and surprise—along with the neutral state, using SA, depth map, and AiF images from IST-EURECOM LFFD.

The SA images can be extracted from the raw images using MATLAB’s LFtoolbox V0.4 [31].

### 4.5. Image Selection from LFFD

We use AiF images and depth maps, each representing a single image per subject, as well as SA images, which constitute a series of images that vary slightly in terms of viewing position.

Consequently, we do not use the entire array of SA images to examine depth information and facial structure. Instead, we employ a selected subset of images from the 15×15 SA image mosaic. Through comprehensive testing of various SA image sets, we have identified that those located on the upward and downward diagonals with a step of 3 are the most effective, as illustrated in Figure 8.

### 4.6. Preprocessing and Data Augmentation

We implement data augmentation techniques on the pre-training dataset to enhance model robustness. Transformations include rotations (−15 to 15), zoom adjustments (−0.15 to 0.15), brightness changes (0.6 to 1.2), shear modifications (−0.15 to 0.15), and horizontal flipping. These adjustments train the model to handle variable angles, scales, lighting conditions, and expressions. We apply the ’Nearest’ fill_mode to maintain image quality during transformations. For both AiF and SA images, we use the Yolo Face algorithm [32] for cropping to 100×75 pixels while maintaining the aspect ratio, before proceeding with training and testing.

### 4.7. Hyperparameters

For our models, we use an input resolution of 100×75 pixels, training for 1000 epochs with a batch size of 16. The networks are compiled using categorical cross-entropy as the loss function, the Adam optimizer, and accuracy as the performance metric.

During training, we employ a ModelCheckpoint callback that focuses on the performance of the final output layer, saving the best weights based on maximum validation accuracy. This ensures the retention of the most effective weights.

Additionally, an EarlyStopping callback is used to monitor the final validation output, with a patience setting of 10 epochs. This stops training if no improvement is observed in the target metric for the specified number of epochs, preventing overfitting and reducing computational cost.

Lastly, the ReduceLROnPlateau callback adjusts the learning rate based on performance of the final output validation. It decrements the learning rate by a factor of 0.5 upon no metric improvement, with a patience of 5 epochs and a minimum learning rate of $1\times 10^{-6}$. This approach aids in refining the model, making more precise weight adjustments as it converges to the optimal solution, thus enhancing overall model performances.

### 4.8. Protocols for Model Evaluation

To compare our proposed method to state-of-the-art techniques, our model is evaluated using two distinct protocols: subject-specific evaluation and subject-independent evaluation.

**Subject-specific evaluation (SSE)**. This method uses data from the first session to train the model and data from the second session for testing and vice versa. The average of the two test results provides insights into the model’s consistency and reliability over time for the same individuals. It tests the model’s ability to generalize across different times for the same person.**Subject-independent evaluation (SIE)**. This uses a Leave-Ten-Subjects-Out Cross-Validation approach. The model is trained on 90% of subjects and tested on the remaining 10%. This is repeated until all subjects have been used for testing, providing an average score that reflects how well the model can predict emotions on people it has never seen before, highlighting its potential real-world effectiveness and adaptability.

These protocols offer a comprehensive assessment of the model’s performance, reliability, and applicability in varied real-world scenarios.

## 5. Evaluation Results and Comprehensive Discussion

In this section, we investigate the outputs derived from the pre-training phase of the EfficientNetV2-S model. We compare the performances of various fusion models against the individual results obtained from processing SA, AiF, and depth map images separately.

### 5.1. Pre-Training Model

In the initial phase of our study, as detailed in Section 3.1, the EfficientNetV2-S exhibited varying levels of accuracy across the CK+48, FER2013, and AffectNet databases, with respective accuracies of 94.87%, 66.51%, and 76.46%. These variations reflect the inherent complexities and diversities within each database, while also highlighting the adaptability and sensitivity of our model to different data characteristics.

Upon implementing Model_1, which processes AiF images, as delineated in Section 4.1, and pre-training on these three databases for emotion recognition under SSE, we noted a marked uniform enhancement in performance metrics. Notably, AffectNet demonstrated the most significant improvement, achieving an average accuracy of 88.38% with a standard deviation (STD) of 8.18%, as illustrated in Table 1.

The model’s high accuracy and reduced STD, when pre-trained on AffectNet, highlight its robustness, consistency, and reliability in emotional recognition tasks across diverse emotional states. AffectNet, with its significantly larger volume and more extensive variety of emotional expressions compared with CK+48 and FER2013, has been instrumental in achieving these results. The ability to handle and learn from such a large, diverse dataset underscores the model’s adaptability and the distinct advantages offered by AffectNet’s comprehensive data in enhancing precision, improving performance and generalization capabilities.

To extend our results, we generated depth maps from the RGB images of AffectNet using the Depth Anything model. The model achieved a score of **81.74%**.

The contribution of the synthesized depth images to FER is significant. Compared with a score of 76.46% obtained using only RGB images, the use of depth maps has considerably improved performance. Depth maps provide additional information about the three-dimensional structure of faces, which is crucial for accurately identifying facial expressions, especially under varying pose and lighting conditions.

By pre-training the CNN on both RGB and depth images, we were able to create a more robust and generalizable model. This approach enables a better understanding of the nuances of emotional expressions by incorporating three-dimensional information, which is not always evident in two-dimensional images. The combination of RGB data and depth maps enriches the model, allowing it to discern subtle details of facial expressions, thus improving the accuracy and reliability of emotion recognition.

### 5.2. Results of Unimodal Architectures

We tested the unimodal models using SSE and SIE protocols on the LFFD dataset. Table 2 represents the performance of the models under an SSE protocol.

The model by Sepas-Moghaddam et al., which uses SA images, achieved an average accuracy of 87.62% with an STD of 5.41%. In comparison, our Model_3, also using SA images, achieved a slightly higher average accuracy of 88.13% with a higher STD of 7.42%. This demonstrates that our model not only matches but slightly surpasses the performance of the Sepas-Moghaddam et al. model in terms of accuracy, while still maintaining a robust performance across various emotional states.

Our Model_1 with AiF images achieved the highest average accuracy among our models, with an accuracy of 88.38% and an STD of 8.18%. This model excelled particularly in recognizing the “Happy” (96.50%), “Neutral” (90%), and Surprise (90%) emotions, outperforming all other models in these categories.

Model_2, which uses depth images, performed the poorest among all the models, with an average accuracy of 42.13% and an STD of 8.17%. This model struggled significantly with recognizing all the emotions. This suboptimal performance can be attributed to the inadequate calibration of the LFFD dataset, where the depth range was not properly adjusted. Additionally, our model was pre-trained on synthesized depth images, which, although it improved the score to some extent, was insufficient to extract the detailed information stored in the LF depth maps necessary for effective emotion recognition.

Model_3, which uses SA images, demonstrated strong performance, achieving an average accuracy of 88.13% and an STD of 7.42%. This model showed strong performance across all emotions, particularly in recognizing the Angry (80.50%), Happy (94.50%), and Surprise (91%) emotions, making it comparable with the AiF image model.

Overall, Model_1 using AiF images demonstrated the highest overall performance in terms of average accuracy and emotional recognition, particularly excelling in the recognition of Happy, Neutral, and Surprise emotions. Model_3 with SA images also showed strong performance, surpassing the state-of-the-art model by Sepas-Moghaddam et al. in terms of average accuracy, although with slightly higher variability. Model_2 with depth images showed that depth information alone is insufficient for robust FER.

As seen in Table 3, our Model_1 with AiF images achieved the highest performance, with an average accuracy of 94.11% and an STD of 4.08%. This model excelled in recognizing the Angry (88.57%), Happy (97.14%), Neutral (92.86%), and Surprise (97.87%) emotions, outperforming all other models in these categories.

On the other hand, Model_2, which employed depth images, recorded an average accuracy of 59.46% with an STD of 7.17%. This model faced considerable difficulties in recognizing all emotions, underscoring the insufficiency of depth information alone for effective emotion recognition.

Model_3, using SA images, also demonstrated strong performance, achieving an average accuracy of 91.88% and an STD of 3.25%. This model was particularly effective in recognizing the Happy (95%), Neutral (94.17%), and Surprise (91.67%) emotions, highlighting its robustness and reliability. The model developed by Sepas-Moghaddam et al. achieved an average accuracy of 80.37% with an STD of 9.03%. Despite performing reasonably well, it was outperformed by Model_3.

Overall, Model_1 with AiF images demonstrated the best performance across both protocols, highlighting the effectiveness of all-in-focus images for unimodal emotion recognition tasks. Model_3 with SA images also showed strong performance, further emphasizing the potential of SA images. Conversely, Model_2 with depth images indicated the limitations of relying solely on depth information for such tasks.

### 5.3. Results for Fusion Strategies

In this section, we compare decision-level versus feature-level fusion methods, elaborated in Section 4.2. Table 4 summarizes the fusion approaches using an EfficientNetV2-S model pre-trained on AffectNet, following an SSE protocol. Our objective is to enhance the FER process by leveraging detailed data from LF camera technology, such as AiF and SA images, aiming to deepen our understanding of how different fusion strategies affect computational emotion analysis.

We employed various fusion strategies, including sum, maximum, multiply, average, and concatenation, each with unique advantages for enhancing the FER process.

Decision-level fusion methods showed promising outcomes. The sum fusion approach, which integrates complementary data to enhance robustness, achieved an average accuracy of 87.00% with an STD of 2.74%. This method’s ability to consolidate diverse information sources likely contributed to its consistent performance. Similarly, the maximum fusion method, which prioritizes the most salient features, resulted in an average accuracy of 86.38% with an STD of 4.77%. The multiply fusion method, emphasizing commonalities across inputs, achieved an accuracy of 86.25% but exhibited higher variability with an STD of 6.41%.

Among the decision-level fusion strategies, the simple average method, which balances the inputs to ensure data consistency, stood out with an average accuracy of 87.88% and an STD of 7.33%. This approach’s balanced handling of input data may account for its superior performance across a range of facial expressions.

Feature-level fusion methods exhibited varied performances. The sum fusion method at the feature level achieved an average accuracy of 86.88% with an STD of 2.02%, indicating its consistent performance. The maximum approach yielded an average accuracy of 86.13%, but with a higher STD of 5.59%, suggesting greater variability. The multiply method resulted in an average accuracy of 85.7% with an STD of 5.17%, while the concatenation method showed an average accuracy of 85.50% with an STD of 6.28%. The simple average approach achieved an accuracy of 86.38% with an STD of 7.33%, indicating a balanced yet variable performance.

The use of LF camera technology, which captures both AiF and SA images, significantly enriched the data available for FER. This technology proved particularly beneficial for decision-level fusion strategies. The average approach, which achieved an average FER score of 87.88%, exemplifies how leveraging detailed LF data can enhance the accuracy and robustness of emotion recognition models. The superior performance of the decision-level fusion methods underscores the importance of integrating diverse data sources to capture the nuances of facial expressions effectively.

Overall, the results indicate that while all fusion strategies can effectively leverage varied data, decision-level fusion, particularly using the simple average method, offers a balanced performance across different emotional states. The inclusion of LF camera data further enhances the capability of these models, demonstrating significant improvements in accuracy and reliability. These findings highlight the potential of advanced fusion techniques and sophisticated imaging technologies in advancing computational emotion analysis.

### 5.4. Benefits of Multimodal Information

Using an SSE protocol on the LFFD dataset, we evaluated the performances of the multimodal models. Table 5 summarizes the results.

Model_4, which integrates SA and depth images, achieved an average accuracy of 86.00% with an STD of 8.00%. This model demonstrated strong performance in recognizing the Angry (75.00%), Happy (88.00%), Neutral (87.00%), and Surprise (94.00%) emotions. Despite its robust performance, Model_4 exhibited slightly less consistency compared with other multimodal combinations.

Model_5, which combines SA and AiF images, demonstrated an average accuracy of 87.50% with an STD of 7.33%. This model excelled in recognizing the Angry (77.00%), Happy (93.00%), Neutral (91.00%), and Surprise (90.50%) emotions. This combination showed a balanced performance across different emotional states.

Model_6, combining AiF and depth images, achieved an average accuracy of 85.88% with an STD of 9.50%. This model performed well in recognizing the Angry (72.00%), Happy (92.50%), Neutral (87.50%), and Surprise (91.50%) emotions. Despite the good performance, the higher standard deviation indicates more variability in its results.

Model_7, which integrates all modalities (SA, AiF, and depth images), demonstrated the best performance, achieving an average accuracy of 90.13% with an STD of 4.95%. This model showed excellent performance across all emotions, particularly in recognizing the Angry (86.50%), Happy (95.00%), Neutral (85.50%), and Surprise (93.50%) emotions. The integration of all modalities provided a comprehensive understanding of facial expressions, leading to the highest accuracy and the lowest variability.

It is important to note that the depth images in the LFFD dataset were not optimally calibrated during capture, leading to a lack of detailed facial structure information. Additionally, the model was pre-trained on synthesized depth images, which did not effectively extract the nuanced information present in the light field depth images. These factors contributed to the lower performance scores observed when using depth images alone. However, the fusion strategy employed in Model_7 effectively leveraged the complementary strengths of SA and AiF images to extract the most pertinent information from the depth modality. By relying on the strengths of the other modalities, Model_7 was able to mitigate the limitations of the depth images, thereby achieving superior overall performance.

Compared with the unimodal models, the multimodal fusion models demonstrated significant improvements in performance. Model_1, which used AiF images alone, achieved the highest accuracy among unimodal models with an average of 88.38%. However, the multimodal Model_7 surpassed this, achieving an accuracy of 90.13%. This demonstrates the substantial benefit of combining multiple modalities, as it allows the model to draw on a richer set of features and improve its recognition capabilities.

Otherwise, we evaluated the multimodal models using an SIE protocol on the LFFD dataset. Table 6 represents the performance of these models.

Model_4 achieved an average accuracy of 90.18% with an STD of 6.26%. This model demonstrated robust performance, particularly in recognizing Angry (85.71%), Happy (95.71%), Neutral (78.57%), and Surprise (89.29%) emotions. However, the higher variability indicates that the combination of SA and depth images is less consistent compared with other multimodal configurations.

Model_5 demonstrated an average accuracy of 95.18% with a standard deviation (STD) of 5.06%. This model excelled in recognizing Angry (87.14%), Happy (99.29%), Neutral (95.71%), and Surprise (98.57%) emotions. The fusion of SA and AiF images provided a well-balanced and robust performance across various emotional states.

Model_6, which combines AiF and depth images, achieved an average accuracy of 86.04% with an STD of 4.02%. This model performed well in recognizing Angry (80.00%), Happy (90.83%), Neutral (88.33%), and Surprise (85.83%) emotions. However, the relatively higher standard deviation suggests more variability in its results, indicating that the combination of AiF and depth images is less consistent compared with other multimodal approaches.

Model_7, which integrates all modalities, demonstrated the best performance, achieving an average accuracy of 93.33% with an STD of 4.92%. This model showed excellent performance across all emotions, particularly in recognizing Angry (86.67%), Happy (100%), Neutral (91.67%), and Surprise (95%) emotions. The integration of all modalities provided a comprehensive understanding of facial expressions, resulting in high accuracy and low variability.

Compared with the unimodal models, the multimodal fusion models demonstrated significant improvements in performance. Model_1, which used AiF images alone, achieved the highest accuracy among unimodal models with an average of 94.11% under an SIE protocol. However, the multimodal Model_5 surpassed this, achieving an accuracy of 95.18%. This demonstrates the substantial benefit of combining multiple modalities, as it allows the model to draw on a richer set of features and improve its recognition capabilities.

In conclusion, Model_5 demonstrated the highest performance and robust emotion recognition across various states, indicating the effectiveness of combining SA and AiF images. Model_7, which integrated all modalities, also performed exceptionally well with low variability, highlighting the benefits of a comprehensive multimodal approach. While Model_4 and Model_6 showed strong performance, their higher variability suggests that certain combinations, such as SA and depth images, may be less consistent. Overall, these findings underscore the substantial benefits of multimodal fusion in enhancing the accuracy and robustness of facial emotion recognition.

Diving deeper into the evaluation under an SIE protocol, we aim to provide a more granular understanding of our model’s performance. In one of the test instances, our approach yielded accuracy scores of 90% for ‘Angry’, 100% for ‘Happiness’, 95% for ‘Neutral’, and 100% for ‘Surprise’, reaching an impressive average accuracy of 96.25%. While these scores are promising, we seek to further dissect the results. To gain insights into the model’s behavior, we showcase images that were incorrectly classified by our model, shedding light on areas where improvement is possible (see Figure 9).

To further our analysis, we engaged 32 individuals to answer a questionnaire to rate on a scale from 1 to 5 for four facial expressions (‘Angry’, ‘Happiness’, ‘Surprise’ and ‘Neutral’), specifically for those three misannotated images. Figure 10 gives the scores for each emotion across the three images, providing a nuanced view of human perception in relation to the model’s misclassifications. This approach allows for a more detailed understanding of the subtleties involved in FER and highlights potential areas for enhancing the model’s accuracy.

Regarding the data illustrated in Figure 10 concerning the three misannotated images, we analyze the feedback provided by the participants based on the graph.

First, the graph likely presents the distribution of scores for each emotion for the three images. This is essential to observe the trends and patterns that emerge from the participants’ ratings.

For Image 1 of Figure 9 with the predicted ‘Angry’ and expected ‘Neutral’, we might observe a higher concentration of ratings around the median for ‘Neutral’, as indicated by the average score of 2.75. This central tendency could suggest a general agreement among participants towards a neutral expression, despite the model’s prediction of ‘Angry’. The lower scores for ‘Angry’, ‘Happiness’, and ‘Surprise’ might be spread out or clustered towards the lower end of the scale, indicating less agreement or confidence in these facial expressions for Image 1.

Moving to Image 2 of Figure 9, predicted ‘Neutral’ but expected ‘Angry’, the graph might show a more even distribution of scores for ‘Angry’ and ‘Neutral’, reflecting the closer average scores (1.72 for ‘Angry’ and 2.66 for ‘Neutral’). This could suggest a divided perception among participants, with some leaning towards a neutral expression and others perceiving anger. The distribution of ‘Happy’ and ‘Surprise’ scores might again show lesser variance and lower averages, reinforcing the idea that these were not the dominant perceived emotions for this image.

Finally, for Image 3 of Figure 9, with a prediction of ‘Angry’ but an expectation of ‘Neutral’, the graph might show a pronounced peak or a higher average for ‘Neutral’ at 3.03, indicating a strong consensus towards a neutral expression among participants. The ‘Angry’ score, while lower, might show a broader spread or a secondary peak, reflecting a significant minority of participants who align with the model’s prediction. The scores for ‘Happy’ and ‘Surprise’ might remain consistently low, as with the other images.

In summary, the graph in Figure 10 provides a visual representation of these distributions and tendencies, offering a clearer picture of the collective human judgment versus the model’s predictions. It is important to note that the dataset contains simulated images, and in some cases, the facial expressions of certain subjects are ambiguous. This ambiguity may be a contributing factor to the discrepancies observed between the model’s predictions and human perception. By dissecting these patterns, we can better understand where the model aligns or diverges from human perception and how emotion recognition might be fine-tuned for improved accuracy and understanding.

## 6. Conclusions

This article aims to study the contribution of different modalities obtained with a light field (LF) camera for facial expression recognition (FER). We present the optical imaging system used and the variety of information it provides. We explore the contribution of each modality separately and test various fusion strategies to combine these modalities effectively.

Our findings demonstrate the potential of using light field cameras to enhance facial expression recognition through multimodal fusion strategies. Our experiments show that combining sub-aperture, all-in-focus, and depth images significantly improves FER accuracy and robustness. The decision-level fusion, particularly with average weights, achieved the highest performance, underscoring the importance of integrating varied data sources.

Compared with unimodal models, the multimodal fusion models exhibited superior performance, highlighting the advantages of using comprehensive data from LF cameras. Notably, our multimodal model, which integrates all three modalities, achieved the best results, with an accuracy of 90.13% ± 4.95% under the subject-specific evaluation protocol and 93.33% ± 4.92% under the subject-independent evaluation protocol. These results indicate the effectiveness of our fusion strategy in capturing nuanced emotional expressions.

Furthermore, the observed misclassification in some instances can be attributed to the difficulty certain subjects encountered in simulating the required emotions. This emphasizes the need for training datasets to include more naturalistic emotional expressions.

Looking ahead, we plan to develop a new dataset using a plenoptic 2.0 camera, focusing on the six basic emotions and the neutral state. This dataset is calibrated to optimize the depth range for facial images, allowing for more precise FER. This future work aims to create a robust system capable of recognizing facial expressions using the rich data provided by light field cameras, further advancing the field of affective computing.

## Figures and Tables

**Figure 1 sensors-24-05724-f001:**
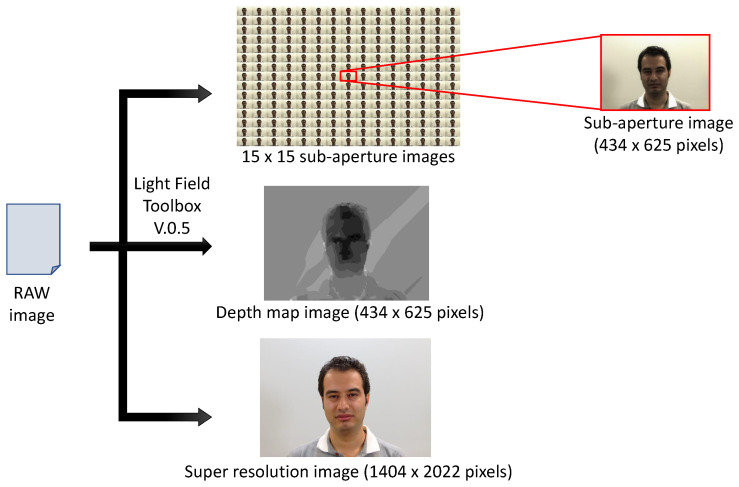
The IST-EURECOM Light Field Face Database (LFFD) [4] showcases sub-aperture, all-in-focus, and depth map images, illustrating the diversity and multimodality of single-sensor imaging.

**Figure 2 sensors-24-05724-f002:**
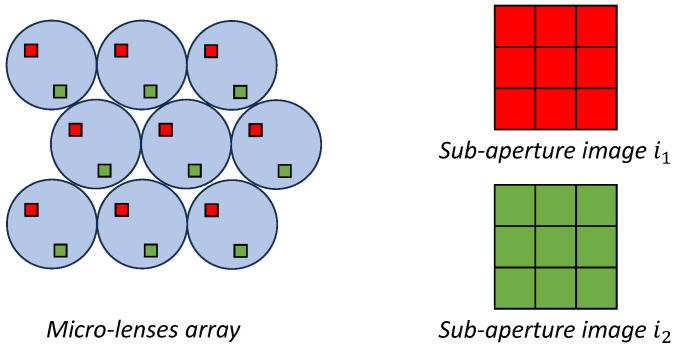
A schematic representation of sub-aperture image extraction.

**Figure 3 sensors-24-05724-f003:**
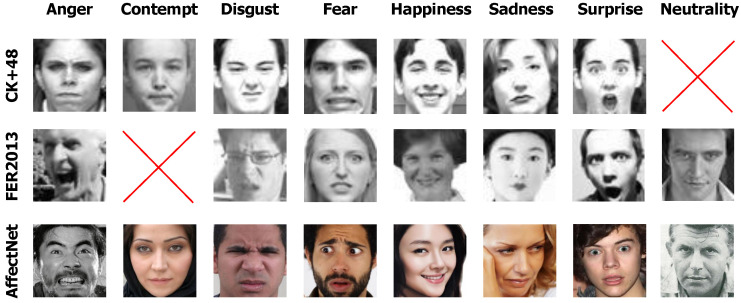
Sample emotion representations across datasets: FER2013, CK+48, and AffectNet.

**Figure 4 sensors-24-05724-f004:**
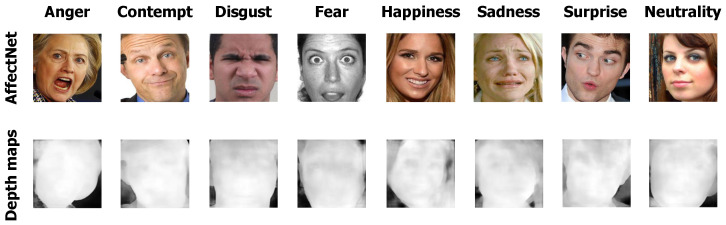
The AffectNet dataset provides diverse sample emotion representations, complemented by synthesized depth maps, illustrating the range and complexity of data used in the study.

**Figure 5 sensors-24-05724-f005:**
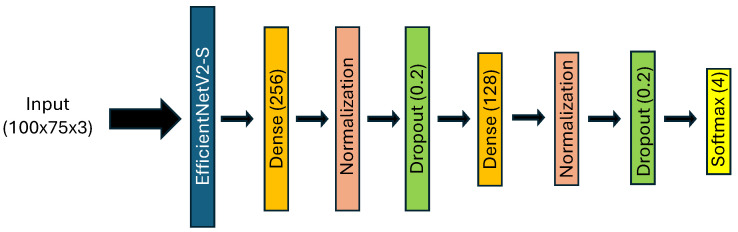
Architecture of Model_1 and Model_2 for AiF and depth images, respectively, using EfficientNetV2-S.

**Figure 6 sensors-24-05724-f006:**
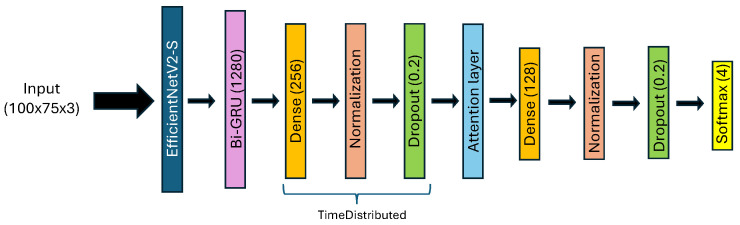
Architecture of Model_3 for SA images, using EfficientNetV2-S.

**Figure 7 sensors-24-05724-f007:**
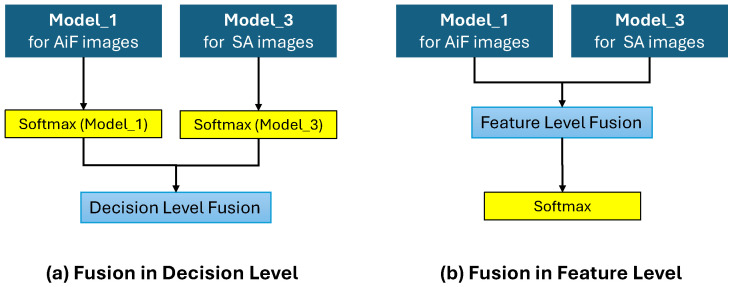
A comparative diagram of decision-level and feature-level fusion strategies.

**Figure 8 sensors-24-05724-f008:**
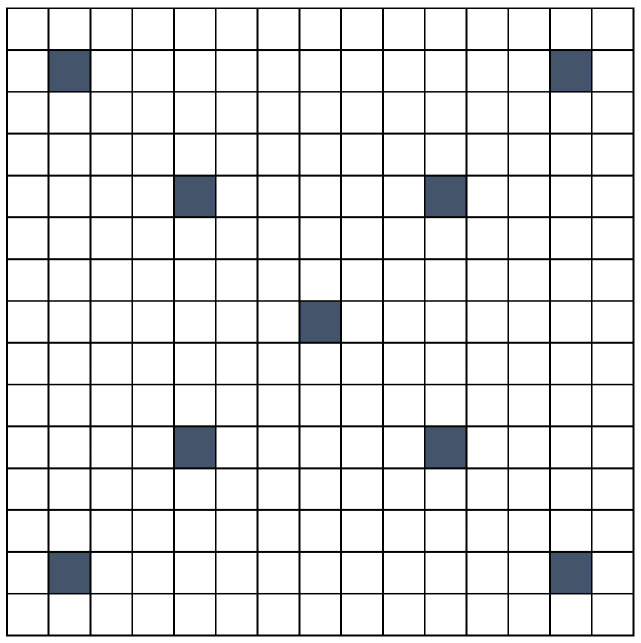
Used set of SA images.

**Figure 9 sensors-24-05724-f009:**
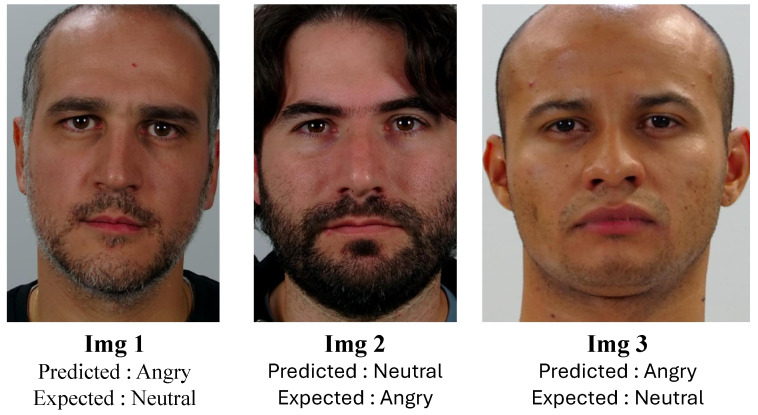
Misannotated images by the fusion model for one of the SIE protocol test instances.

**Figure 10 sensors-24-05724-f010:**
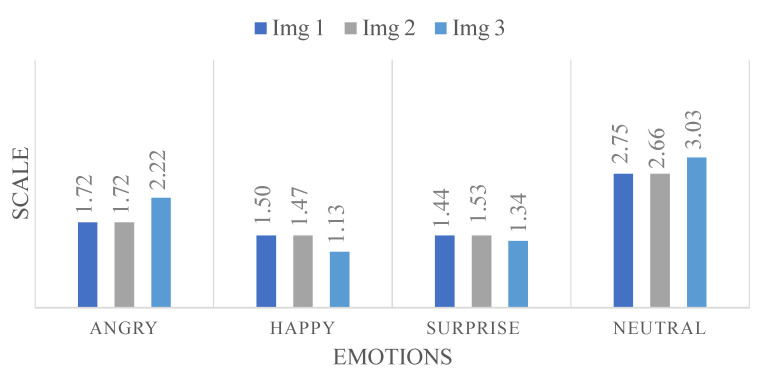
Participant rating distribution for misannotated emotion predictions.

**Table 1 sensors-24-05724-t001:** Performance metrics of Model_1 on CK+48, FER2013, and AffectNet under SSE protocol with LFFD.

Emotions	CK+48 (%)	FER2013 (%)	AffectNet (%)
Angry (%)	76.50	76.00	**77.00**
Happy (%)	92.50	85.50	**96.50**
Neutral (%)	82.00	80.00	**90.00**
Surprise (%)	85.00	90.00	**90.00**
Avg(%) ± STD (%)	84.00 ± 6.67	82.88 ± 6.14	**88.38** ± **8.18**

**Table 2 sensors-24-05724-t002:** Performance of different unimodal models on the LFFD dataset under SSE protocol and comparison with the state of the art.

	Angry (%)	Happy (%)	Neutral (%)	Surprise (%)	Avg (%) ± SD (%)
Model of [11] (SA images)	**88**	94	81	87.50	87.62 ± **5.41**
Model_1 with AiF images	77	**96.50**	**90**	**90**	**88.38** ± 8.18
Model_2 with Depth images	43.50	52.50	33	39.50	42.13 ± 8.17
model_3 with SA images	80.50	94.50	86.50	91	88.13 ± 7.42

**Table 3 sensors-24-05724-t003:** Performance of different unimodal models on the LFFD dataset under SIE protocol and comparison with the state of the art.

	Angry (%)	Happy (%)	Neutral (%)	Surprise (%)	Avg (%) ± SD (%)
Model of [11] (SA images)	80.50	86	71.50	83.50	80 ± 9.03
Model_1 with AiF images	**88.57**	**97.14**	92.86	**97.87**	**94.11** ± 4.08
Model_2 with Depth images	64.29	67.14	48.57	57.86	59.46 ± 7.17
model_3 with SA images	86.67	95	**94.17**	91.67	91.88 ± **3.25**

**Table 4 sensors-24-05724-t004:** Comparative analysis of decision-level fusion and feature-level fusion techniques on Model_5.

**Decision-Level Fusion**					
**Fusion Type**	**Angry (%)**	**Happy (%)**	**Neutral (%)**	**Surprise (%)**	**Avg (%) ± SD (%)**
Sum	**89.00**	87.50	88.50	83.00	87.00 ± 2.74
Maximum	80.00	85.50	90.50	89.50	86.38 ± 4.77
Multiply	83.50	80	86.50	95	86.25 ± 6.41
Average	77.00	93.00	91.00	**90.50**	**87.88** ± 7.33
**Feature-Level Fusion**					
**Fusion Type**	**Angry (%)**	**Happy (%)**	**Neutral (%)**	**Surprise (%)**	**Avg (%) ± SD (%)**
Sum	87.00	84.00	88.50	88.00	86.88 ± **2.02**
Maximum	79.00	84.50	**91.50**	89.50	86.13 ± 5.59
Multiply	84.50	79.00	89.00	**90.50**	85.75 ± 5.17
Concatenation	85.00	**94.50**	80.50	82.00	85.50 ± 6.28
Average	78.00	91.00	90	86.50	86.38 ± 7.33

**Table 5 sensors-24-05724-t005:** Performance of different multimodal models on the LFFD Dataset under SSE protocol.

	Angry (%)	Happy (%)	Neutral (%)	Surprise (%)	Avg (%) ± SD (%)
Model_4 using SA + Depth images	75.00	88.00	87.00	**94.00**	86.00 ± 8.00
Model_5 using SA + AiF images	77.00	93.00	**91.00**	90.50	87.50 ± 7.33
Model_6 using AiF + Depth images	72.00	92.50	87.50	91.50	85.88 ± 9.50
Model_7 using all modalities	**86.50**	**95.00**	85.50	93.50	**90.13** ± **4.95**

**Table 6 sensors-24-05724-t006:** Performance of different multimodal models on the LFFD Dataset under SIE protocol.

	Angry (%)	Happy (%)	Neutral (%)	Surprise (%)	Avg (%) ± SD (%)
Model_4 with SA + Depth images	85.71	95.71	78.57	89.29	90.18 ± 6.26
Model_5 with SA + AiF images	**87.14**	**99.29**	**95.71**	**98.57**	**95.18** ± 5.06
Model_6 with AiF + Depth images	80.00	90.83	88.33	85.83	86.04 ± **4.02**
Model_7 with all modalities	86.67	100	91.67	95	93.33 ± 4.92

## Data Availability

Restrictions apply to the availability of these data. Data were obtained from EURECOM, a research center in digital science, and are available at https://lffd.eurecom.fr/ with the permission of A. Sepas-Moghaddam.

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
