# Peer review of "Enhancing Facial Expression Recognition through Light Field Cameras"

_sensors, 2024, doi:10.3390/s24175724_

Round 1
Reviewer 1 Report
Comments and Suggestions for Authors
The manuscript investigated a multimodal facial expression recognition method based on light field camera. The study compared the recognition results of unimodal models and multimodal fusion under different fusion strategies, as well as different fusion modes of sub-aperture,full-in-focus, and depth maps from the light field camera. The study also investigated the recognition capabilities of different fusion modes under two evaluation methods. The topic is interesting and useful.
1. Figure 8 is somewhat redundant, and the text description is already clear.
2. The model with depth maps did not yield good results. What is the fundamental reason for this? Is it due to the composition of the dataset? Is it possible to directly use LF to obtain depth maps and construct the dataset?
3. As shown in Table 6, under the SIE protocol, Model 5 performs better than Model 7. What does this indicate? Does it mean that depth information may not be necessary?
4. How were the test instances described in lines 509-513 obtained? No relevant data was found in Tables 5 and 6.
5. Does “protocol 2” in Figure 10 refer to SSE or SIE?
6. The discussion on the issue of misannotated images does not provide a clear conclusion. In other words, it has not been clearly described yet what problem related to facial expression recognition with light field cameras is being addressed.
Author Response
Comments 1 : Figure 8 is somewhat redundant, and the text description is already clear.Response 1 : We acknowledge your observation and agree that Figure 8 may be redundant given the clarity of the accompanying text. As a result, we have decided to remove Figure 8 from the article to streamline the presentation and avoid unnecessary repetition.
Comments 2 : The model with depth maps did not yield good results. What is the fundamental reason for this? Is it due to the composition of the dataset? Is it possible to directly use LF to obtain depth maps and construct the dataset?
Response 2 : The suboptimal performance of the model using depth maps can be attributed to two primary factors. First, the LFFD dataset was captured without adjusting the depth field before image acquisition. This limitation stems from the use of Lytro cameras, which belong to the first generation of plenoptic cameras and do not support depth field adjustment post-capture. In contrast, second-generation light field cameras like those from Raytrix allow for such adjustments, potentially enhancing the quality and utility of the depth maps. Second, the model was pre-trained on synthesized depth images, which may not fully capture the complexity and nuances of real-world depth data. This discrepancy likely contributed to the model's difficulty in effectively leveraging depth information. To address these issues, we are currently developing a new dataset consisting of 166 subjects, captured with a Raytrix camera, including the six basic emotions and a neutral state. This new dataset will allow us to more thoroughly explore and compare the efficacy of depth information in facial expression recognition in future studies. Comments 3 : As shown in Table 6, under the SIE protocol, Model 5 performs better than Model 7. What does this indicate? Does it mean that depth information may not be necessary?
Response 3 : The observed performance of Model 7, which incorporates depth maps, suggests that the inclusion of this modality may negatively impact accuracy, primarily due to the challenges our model faces in effectively recognizing depth information.This indicates that the current method of integrating depth maps might introduce conflicting information, thereby reducing the overall performance of the fusion model. Our forthcoming efforts will focus on developing a more sophisticated fusion model that can intelligently manage or disregard conflicting data, potentially improving accuracy. However, it is too early to dismiss the value of depth maps based solely on the results from the LFFD dataset. The utility of depth information in facial expression recognition is an open question that we intend to explore further with our new dataset, which is nearing completion. Comments 4 : How were the test instances described in lines 509-513 obtained? No relevant data was found in Tables 5 and 6.
Response 4 : The test instances described in lines 509-513 were actually obtained under the SIE (Subject Independent Evaluation) protocol, not the SSE protocol. We apologize for this mistake. The SIE protocol involves using 90% of the subjects for training and 10% for testing, rotating the subjects to create different test groups. For the analysis presented in those lines, we focused on one specific group from this protocol to conduct a detailed study. The results mentioned in the text represent the performance on this specific group, highlighting the model’s accuracy for different emotions. In Table 6, we presented the average accuracy across all groups, which is why the specific results from this group were not separately detailed in the tables.
Comments 5 : Does “protocol 2” in Figure 10 refer to SSE or SIE?
Response 5 : “Protocol 2” in Figure 10 refers to the SIE (Subject Independent Evaluation) protocol. We apologize for the ambiguity and have revised the figure and corresponding text to clearly indicate this. Comments 6 : The discussion on the issue of misannotated images does not provide a clear conclusion. In other words, it has not been clearly described yet what problem related to facial expression recognition with light field cameras is being addressed.
Response 6 : The connection to light field cameras is crucial, as the data captured by these cameras has enabled our model to achieve performance levels that surpass the current state of the art. However, it is evident that the model's accuracy could have been further enhanced if not for the presence of inaccurate or ambiguous emotional simulations by certain subjects within the dataset. We have now addressed this issue in more detail in the conclusion of the section.
Reviewer 2 Report
Comments and Suggestions for Authors
The approach of using the LF camera for facial expression recognition has potential for the future. However, the integration of SA image, Ai F image, and depth map that the author claims in the paper is not unique in its integration method. The results do not show the effectiveness of the depth map, and the conclusion seems to be that the SA and AiF images are sufficient. Judging from the depth image, it appears that the depth component is not sufficient to discriminate facial expressions. Although depth images are considered effective, it would be better to use a method that detects effective depth information, such as a ToF camera.
Finally, the description at the end of chapter 5 was very confusing. It is doubtful that this description is necessary for this paper.
Comments on the Quality of English LanguageIt's better not to begin in the first person.
Author Response
Dear Reviewer,
Thank you for your constructive feedback on our manuscript. We would like to clarify a key point regarding the use of depth maps in our study. The LFFD database we initially used did not have the camera configured for an optimal depth range to capture detailed facial features, resulting in depth maps that did not provide sufficient information for effective facial expression discrimination. To address this, we are currently finalizing the construction of our own database using a Raytrix plenoptic 2.0 camera to study the six basic emotions plus the neutral state. With this new setup, the depth maps obtained have shown a very high accuracy score so far. We believe that this approach will overcome the limitations observed with the LFFD database and provide more reliable data for facial expression recognition. We appreciate your valuable suggestions and hope these clarifications meet your expectations.
Reviewer 3 Report
Comments and Suggestions for Authors
It is not always possible to determine what a person is feeling by their appearance. People with different psychotypes express their emotions in different ways. In such a situation, each individual method is not always effective. A comprehensive solution can significantly increase the accuracy of determining the psychological state of a person. To improve the accuracy of emotion recognition, the authors of the paper combined three modalities: sub-aperture images, depth maps and all-in-focus images. The authors obtained encouraging results.
However, I was left with questions.
There are seven different emotions in the datasets used. Why are only three used for analysis?
The authors use the notion of accuracy and performance. I would like to see in the article how they calculate these concepts.
The structure of the datasets is not specified in the text of the paper. The authors should have analyzed the classification results in more depth. Given the huge difference in the results of processing different datasets, it would be very interesting to look at the Confusion Matrix and ROC curve.
The authors did not explain why EfficientNetV2-S is used as the basis for CNN. The argument that it is known for its performance, efficiency and scalability is not very convincing. Given the huge variation in classification results for different datasets, the authors should have experimented with other neural networks. It would be interesting to see how capsule networks work with data. An example can be seen here: DOI: 10.1007/s11416-023-00500-2
The main difference between Model 1 and Model 2 architectures and Model 3 architectures is bidirectional gated recurrent units (Bi-GRUs). Authors should explain in detail how they are organized.
Citation: «This model leverages advanced deep learning techniques to accurately predict depth information from 2D images, making it highly effective for generating high-quality depth maps that capture intricate details and variations in facial structures.» (176-179)
Questions I would like to have answered: What the leverages advanced deep learning techniques? What does it mean to be very effective?
The last few pages of the manuscript are an enumeration of results. I recommend that the authors present these data in a more visual way.
Author Response
Comments 1 : There are seven different emotions in the datasets used. Why are only three used for analysis? Response 1 : The dataset we used actually contains only three of the six basic emotions, in addition to the neutral state. The dataset also includes various images with changes in lighting, occlusions, and profile views, which are primarily intended for facial recognition rather than facial expression analysis. These additional images were not suitable for our study. Given this limitation, our analysis was confined to the emotions available in the dataset. We have revised the text to better explain the content and limitations of the dataset to avoid any confusion. Comments 2 : The authors use the notion of accuracy and performance. I would like to see in the article how they calculate these concepts.
Response 3 : In our deep learning model, the concepts of accuracy and performance are derived from the output of the Softmax layer, which is the final layer of our neural network. The Softmax layer converts the raw logits from the network into a probability distribution over the possible classes (in this case, the different facial expressions). Accuracy is calculated by comparing the predicted class (the class with the highest probability from the Softmax output) to the true class for each input. The accuracy is then computed as the ratio of the number of correct predictions to the total number of predictions. Comments 3 : The structure of the datasets is not specified in the text of the paper. The authors should have analyzed the classification results in more depth. Given the huge difference in the results of processing different datasets, it would be very interesting to look at the Confusion Matrix and ROC curve.
Response 3 : Thank you for your feedback. The dataset used in our study is the IST-EURECOM Light Field Face Database. We have detailed the distribution and characteristics of this dataset in Section 4.4 of the article, where we have highlighted the most crucial aspects relevant to our analysis. Regarding the confusion matrix and ROC curve, our two evaluation protocols involve repeating processes across multiple iterations and averaging the results. Due to the nature of these protocols, it is challenging to provide detailed confusion matrices and ROC curves for each individual test. However, we have focused on presenting the average results to provide a comprehensive understanding of the model’s overall performance. Comments 4 : The authors did not explain why EfficientNetV2-S is used as the basis for CNN. The argument that it is known for its performance, efficiency and scalability is not very convincing. Given the huge variation in classification results for different datasets, the authors should have experimented with other neural networks. It would be interesting to see how capsule networks work with data. An example can be seen here: DOI: 10.1007/s11416-023-00500-2
Response 4 : EfficientNetV2-S was initially selected due to its well-documented balance of performance, efficiency, and scalability, which we considered suitable for our application. The variations in the results can be attributed primarily to the depth images, which were captured without the necessary pre-capture depth range adjustment in the Lytro camera used by the authors of the IST-EURECOM dataset. This lack of adjustment impacted the quality of the depth maps and, consequently, the performance of our model. Regarding your suggestion to experiment with capsule networks, it is important to note that the authors of the IST-EURECOM dataset have already explored this approach in a previous study. In the work by Sepas-Moghaddam et al., titled "Capsfield: Light field-based face and expression recognition in the wild using capsule routing" (IEEE Transactions on Image Processing 30, 2021: 2627-2642), capsule networks were employed for facial expression recognition. However, their methodology involved grouping the emotions together and comparing these groups primarily against the neutral state, rather than evaluating each emotion individually. To further validate our approach, we are currently developing a new dataset containing all six basic emotions plus the neutral state, captured using Raytrix cameras. This new dataset will allow us to demonstrate the effectiveness of our data fusion method and the stability of our model when using EfficientNetV2-S. Comments 5 : The main difference between Model 1 and Model 2 architectures and Model 3 architectures is bidirectional gated recurrent units (Bi-GRUs). Authors should explain in detail how they are organized.
Response 5 : We introduced the Bi-GRU layer specifically to enhance the extraction of angular information from the sub-aperture images. Sub-aperture images, which capture the scene from slightly different perspectives, contain valuable angular data that can improve the model’s understanding of depth and structure within the scene. The Bi-GRU operates by processing the sequential data from these sub-aperture images in both forward and backward directions, allowing the model to capture dependencies and relationships across different viewpoints. Comments 6 : Citation: «This model leverages advanced deep learning techniques to accurately predict depth information from 2D images, making it highly effective for generating high-quality depth maps that capture intricate details and variations in facial structures.» (176-179). Questions I would like to have answered: What the leverages advanced deep learning techniques? What does it mean to be very effective?
Reponse 6 : The model referenced in the article is the Depth Anything Model (DAM) as introduced in "Depth Map for FER Dataset". By "leverages advanced deep learning techniques," the authors of DAM are referring to the use of a vision transformer architecture, which is a cutting-edge approach in the field of deep learning. Vision transformers have shown remarkable success in various vision tasks due to their ability to model long-range dependencies and capture global context within images. This model has been trained on a large-scale dataset comprising approximately 63.5 million images, which enhances its capability to generalize and accurately predict depth information from 2D images. The phrase "highly effective" refers to the model’s performance in generating depth maps that are not only accurate but also detailed. These depth maps capture subtle variations in facial structures, which are crucial for tasks like facial expression recognition. The effectiveness is supported by the model’s success in producing high-quality depth estimates that contribute to improved recognition performance when incorporated into facial expression recognition systems. In the article, we have cited the original work on DAM to provide readers with a more in-depth understanding of the techniques used and the performance claims made by the authors of the DAM model. Comments 7 : The last few pages of the article are an enumeration of results. I recommend that the authors present these data in a more visual way.
Response 7 : We appreciate the importance of visual representation in conveying complex results more effectively. We would like to point out that Figure 10 specifically addresses this by presenting the participant rating distribution for misannotated emotion predictions. This figure was designed to visually summarize the key findings and provide a clear illustration of how the predictions deviate from the expected results. We believe that this visual representation, along with the accompanying analysis, helps to make the data more accessible and easier to interpret.
Round 2
Reviewer 1 Report
Comments and Suggestions for Authors
The revision has clarified the concerns, and the changes have helped to improve the paper. The evaluation is convincing.
Reviewer 3 Report
Comments and Suggestions for Authors
Thank you for your comprehensive replies. A very interesting manuscript indeed. I would like to see the results obtained with the plenoptic 2.0 camera in your next manuscripts.